# Association between being Overweight in Young Childhood and during School Age and Puberty

**DOI:** 10.3390/children10050909

**Published:** 2023-05-22

**Authors:** Genki Shinoda, Yudai Nagaoka, Fumihiko Ueno, Naoyuki Kurokawa, Ippei Takahashi, Tomomi Onuma, Aoi Noda, Keiko Murakami, Mami Ishikuro, Taku Obara, Hirohito Metoki, Junichi Sugawara, Shinichi Kuriyama

**Affiliations:** 1Division of Preventive Medicine and Epidemiology, Tohoku Medical Megabank Organization, Tohoku University, 2-1 Seiryo-Machi, Aoba-Ku, Sendai 980-8573, Japan; 2Department of Molecular Epidemiology, Graduate School of Medicine, Tohoku University, 2-1 Seiryo-Machi, Aoba-Ku, Sendai 980-8573, Japan; 3School of Medicine, Tohoku University, 2-1 Seiryo-Machi, Aoba-Ku, Sendai 980-8573, Japan; 4Graduate School of Education, Miyagi University of Education, 149 Aramaki-Aza-Aoba, Aoba-Ku, Sendai 980-0845, Japan; 5Department of Pharmaceutical Sciences, Tohoku University Hospital, 1-1 Seiryo-Machi, Aoba-Ku, Sendai 980-8574, Japan; 6Faculty of Medicine, Tohoku Medical and Pharmaceutical University, 1-15-1 Fukumuro, Miyagino-Ku, Sendai 983-8536, Japan; 7Suzuki Memorial Hospital, 3-5-5, Satonomori, Iwanumashi 989-2481, Japan; 8International Research Institute of Disaster Science, Tohoku University, 468-1 Aramakiaoba, Aoba-Ku, Sendai 980-8572, Japan

**Keywords:** body type, birth weight, young childhood, obesity, overweight, puberty, school age

## Abstract

To examine whether body type at birth, body weight, and obesity in early childhood are associated with overweight/obesity during school age and puberty. Data from maternal and child health handbooks, baby health checkup information, and school physical examination information of participants at birth and three-generation cohort studies were linked. Association between body type and body weight at different time intervals (at birth and at 1.5, 3.5, 6, 11, and 14 years of age) were comprehensively analyzed using a multivariate regression model adjusted for gender, maternal age at childbirth, maternal parity, and maternal body mass index, and drinking and smoking statuses at pregnancy confirmation. Children who are overweight in young childhood had a greater risk of being overweight. Particularly, overweight at one year of age during checkup was associated with overweight at 3.5 years (adjusted odds ratio (aOR), 13.42; 95% confidence interval (CI), 4.46–45.42), 6 years (aOR, 6.94; 95% CI, 1.64–33.46), and 11 years (aOR, 5.22; 95% CI, 1.25–24.79) of age. Therefore, being overweight in young childhood could increase the risk of being overweight and obese during school age and puberty. Early intervention in young childhood may be warranted to prevent obesity during school age and puberty.

## 1. Introduction

Obesity is an important public health concern because it contributes significantly to the major components of metabolic syndrome, carries a high risk of health complications, and is associated with social and economic consequences [1]. The World Health Organization has reported that the worldwide prevalence of overweight and obesity in childhood and puberty (aged 5–19 years old) has risen dramatically from 4% in 1975 to over 18% in 2016, and this increase has occurred in both genders [2]. In adulthood, obesity is a primary risk factor for cardiovascular disease [3], type 2 diabetes [4], cancer [5], psychiatric disorders [6], and other non-communicable diseases [7]. According to a study using data from the 2019 Global Burden of Disease Study, obesity-related disability-adjusted life years increased by 0.48% annually between 2000–2019 and is predicted to increase by 39.8% between 2020–2030 [8].

Obesity in childhood is carried into adulthood, a phenomenon known as “Tracking phenomena” [9]; more specifically, obesity in puberty puts an individual at greater risk of obesity in adulthood [10], and children who are obese in the first grade of elementary school are likely to be obese in the sixth grade [11]; approximately 80% of overweight/obese Japanese elementary school students are overweight/obese in junior high school [12]. Obese children are at an increased risk of morbidities, including diabetes, coronary heart disease, and a range of cancers [13]. These findings highlighted the need for interventions to prevent childhood obesity. Furthermore, maternal body mass index (BMI) [14], drinking alcohol [15], and smoking habits [16] have been reported to be risk factors for obesity in children. Collectively, tracking data during a child’s growth process from the prenatal period is required.

Based on the School Health Statistics (School Health Statistical Survey Report) and the Ministry of Education, Culture, Sports, Science, and Technology, the incidence of obesity among children in Japan has remained stable and has increased slightly over the past decade [17]. One possible explanation is that obesity in young childhood and early childhood persists during school age and beyond. However, in Japan, insufficient data have been accumulated to track individuals from birth to school age. Consequently, even if early interventions are implemented, patients are at risk of premature discontinuation.

Studies linking mother-child health and school health information to explore the correlation between child body type in young childhood and early childhood and obesity during puberty are lacking in Japan. Therefore, this study aimed to examine the association between overweight during young childhood, school age, and puberty.

## 2. Materials and Methods

### 2.1. Study Design

Data were obtained from the Tohoku Medical Megabank Project Birth and Three-Generation Cohort Study (TMM BirThree Cohort Study), as previously described elsewhere [18,19,20]. Pregnant women and their children were recruited between 2013–2017 at obstetric clinics and hospitals in the Miyagi Prefecture, Japan. We worked with various municipal boards of education and schools to collect school checkup information for the participants in this cohort between 2018–2020. Inclusion criteria were children whose parents provided proxy consent for regular school health checkups (school checkups). The children were in grade three of a junior high school between 2018–2020, meaning they were born between 2 April 2003 and 1 April 2006. School checkup data were combined with infant health checkup (infant checkup) data collected from municipal mother-child health-related departments and maternal and child health handbooks collected from parents and guardians. Exclusion criteria were refusal to participate and missing data to define body type (gender, height, body weight, measurement dates, date of birth, and gestational age) at any time. In this study, 1.5 to 3.5 years of age was defined as “ young childhood”, 6 to 11 years old as “school age”, and 14 years old as “puberty”. The study protocol was approved by the Ethics Committee of the Tohoku Medical Megabank Organization and conformed to the provisions of the Declaration of Helsinki. Written informed consent was obtained from all the participants. We received guardian’s consent for children between 10–19 years of age with their own assent or consent and parental consent for children <10 years of age.

### 2.2. Definition of Body Type

Body type was defined by gender, height, body weight, measurement date, date of birth, and gestational age. We used data from the participants at six-time points (at birth, age 18–24 months, 36–47 months, 6 years, 11 years, and 14 years). Body type at birth was categorized by examining the body weight of the neonates relative to their gestational age based on Japanese data (growth standard charts for Japanese children, with mean and standard deviation values based on the 2000 national survey [2016]) [21] as follows: body weight <10th percentile, small for gestational age (SGA); body weight between the 10th and 90th percentiles, appropriate for gestational age; and body weight >90th percentile, large for gestational age (LGA) [22]. After birth, BMI z-scores were calculated using reference data for Japanese individuals [23], and body weight status was classified as follows: overweight, BMI z-score >1; normal weight, BMI z-score between −2 and 1; and thin, BMI z-score < −2.

### 2.3. Confounders

We collected data from the Maternal and Child Health Handbook on maternal age at childbirth, maternal parity, maternal height, body weight, maternal BMI, and drinking and smoking statuses when pregnancy was confirmed.

### 2.4. Statistical Analysis

The Wilcoxon rank-sum, Pearson’s chi-squared, and Fisher’s exact tests were used to examine the association between gender and other characteristics. The Fisher’s exact test and logistic regression analysis were used to examine the association between body type at any time point and subsequent time points. We calculated odds ratios (ORs) and 95% confidence intervals (95% CI) for being overweight at any time point and the risk of being overweight at subsequent time points. All analyses were conducted using R v.4.1.1, and statistical significance was set at *p* < 0.05.

## 3. Results

Figure 1 shows a flow diagram of the present study. The participants were 528 children whose guardians provided consent for the collection of regular school health checkup information. Among them, 7 participants withdrew their consent, 251 had missing data (height, body weight, and date of measurement) at all measurement points (at birth and at 1.5, 3.5, 6, 11, and 14 years of age), and 24 were classified as thin at any time point. In total, 246 participants were included in the analysis.

Table 1 shows the characteristics of the participants according to gender. Of the 246 individuals, 119 were boys and 127 were girls; there were no statistically significant differences between the boys and girls at birth in terms of gestational age (weeks), height, or body weight. In terms of body type at birth, four boys were SGA, 16 were LGA, four were SGA, and 20 were LGA. A slightly higher percentage of boys than girls were categorized as SGA (boys, 3.7% vs. girls, 3.4%). However, these differences were not statistically significant. Children for whom missing data were available at each measurement time point were excluded from the analysis for that time point.

Table 2 shows the characteristics of the mothers according to their children’s gender. The maternal age at birth for boys was slightly younger than the maternal age at birth for girls (24.8 ± 3.4 years for boys and 25.1 ± 3.7 years for girls). However, these differences were not statistically significant. Children for whom missing data were available at each measurement time point were excluded from the analysis for that time point.

No significant associations were observed when comparing the percentage of overweight individuals at subsequent measurement time points according to body type at birth. Being overweight at 18–23 months of age was significantly associated with being overweight at 36–47 months (crude odds ratio [cOR], 8.99; 95% CI, 4.13–20.29), 6 years (cOR, 3.86; 95% CI, 1.53–9.65), and 11 years (cOR, 2.89; 95% CI, 1.20–6.84) of age (Figure 2). This association remained after adjusting for potential confounders; at 36–47 months (aOR, 13.42; 95% CI, 4.46–45.42), 6 years (aOR, 6.94; 95% CI, 1.64–33.46), and 11 years (aOR, 5.22; 95% CI, 1.25–24.79) (Table 3). Being overweight at 36–47 months of age was significantly associated with being overweight at 6 years (cOR, 7.40; 95% CI, 2.99–19.38) and 11 years (cOR, 4.17; 95% CI, 1.77–10.01) of age (Figure 2), which remained the same after adjusting for potential confounders at 6 years (aOR, 9.55; 95% CI, 1.85–61.41) and 11 years (aOR, 3.22; 95% CI, 0.70–15.14) (Table 3).

## 4. Discussion

This study examined the association between body type at birth and overweight in young childhood, school age, and puberty and found no association between body type at birth and body weight at each measurement point. However, being overweight during young childhood was significantly correlated with being overweight at school and during puberty. In particular, being overweight during young childhood was strongly correlated with being overweight among school-aged children. Furthermore, this study suggests that the tracking phenomenon of obesity begins in young childhood (i.e., at 1.5 years of age), which supports the need for early intervention in obesity prevention.

Obesity and overweight in childhood are a putative cause of obesity and overweight in adulthood [24,25]. In particular, for obesity during puberty, the greatest body weight gain occurs between 2–6 years of age, and most children who are obese at that age are obese during adolescence [26]. Additionally, obesity during adolescence is significantly associated with an increased risk of cardiovascular and metabolic diseases, such as type 2 diabetes, and several other diseases in adulthood [27]. Additionally, obesity is associated with the onset of cancer and is a potential cause of the onset of central nervous system tumors, including glioblastoma multiforme [28]. Therefore, our finding suggests that interventions for obesity, not only at school age but also during young childhood or earlier, are important for the prevention of obesity in puberty and adulthood. Our study did not observe any relationship between body weight, body type, gestational age at birth, obesity, and being overweight during puberty. A previous study reported modest associations between birth weight, overweight, and obesity at 15–20 years of age, whereas the influence of BMI at 2–4 and 5–7 years on overweight and obesity at 15–20 years was moderate to strong [29]. However, the association between body weight and body type at birth and the risk of being overweight or obese later in life remains inconclusive [30], and further studies are required to examine this association.

Studies using the so-called “life course data,” that is, including data from the child’s fetal stages up to puberty, have reported the latency of data linkage not only for overweight and obesity but also for various healthcare components [31,32,33,34]. In Japan, maternal, child, and school health information are not maximally utilized because of the different ministries and agencies that administer it. Considering the results of this study, linking the life course data is important for formulating measures to prevent obesity during puberty. As obesity in young childhood and childhood impairs body weight loss response to dietary treatments in adulthood, addressing obesity throughout early life may improve the effectiveness of body weight loss interventions in adulthood [35]. This finding highlights the importance of body weight maintenance interventions in infants. Furthermore, obesity is not just a problem for children and adults. With the increasing number of elderly people worldwide, obesity is an important risk factor affecting health in old age. In particular, dementia, including Alzheimer’s disease, is a factor that greatly reduces activities of daily living in old age. However, there is a fragmented understanding of the mechanisms connecting obesity and aging [36]. Therefore, the life course data, including not only childhood data but also adulthood and old age data, will help with proper weight management and obesity prevention.

The strength of this study is its use of existing real-world data, such as maternal and child health handbooks, baby health checkup information, and school physical examination information. However, this study did not include factors related to eating habits or physical activity. Factors of overweight and obesity at school age and puberty include eating habits [37] and physical activity [38], which are also included in obesity prevention programs [39,40]. Additionally, a relationship between nutritional status and being overweight in young childhood has been suggested [41,42], and it is important to consider lifestyle factors, such as diet and physical activity, when investigating childhood obesity. Moreover, the lifestyle of the mother during pregnancy has also been reported to be a risk factor for obesity in children [43,44]. It may be important to include in the analysis environmental factors surrounding the mother and child during the perinatal period, thereby utilizing information from the Maternal and Child Health Handbook, to prevent obesity in the fetal period and thereafter, in children. Young childhood and school age are important periods in which the basis of a child’s lifestyle is formed; therefore, various assessments and interventions based on the linkage of maternal and child health information and school-age information may be important. It is important to follow up with children with obesity and investigate their prognosis in the future. The determination of obesity in our study was based exclusively on height and body weight (i.e., physical indicators), with no examination of body fat. Additionally, the BMI z-scores were calculated with reference to the standard population. Therefore, especially for boys in puberty, it is possible that individuals who were muscular were assigned higher BMI values and categorized as obese. Therefore, the results of the present study should be interpreted with caution.

## 5. Conclusions

This study first linked public information, such as maternal and child health handbooks, baby health checkup information, and school physical examination information, to continuously track the changes in body weight and size from birth to puberty. The findings of our study suggest that among Japanese children, being overweight in young childhood may be associated with being overweight at school age and during puberty. Based on the above results, the prevention of being overweight at school age and puberty requires intervention against being overweight in young childhood, especially during the 1.5-year-old checkup period.

## Figures and Tables

**Figure 1 children-10-00909-f001:**
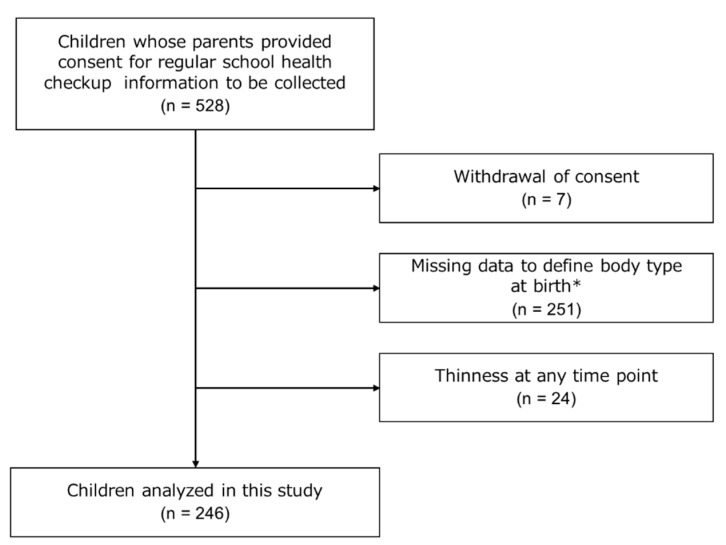
Flow chart of study participants. * Gender, height, body weight, measurement date, date of birth, and gestational age.

**Figure 2 children-10-00909-f002:**
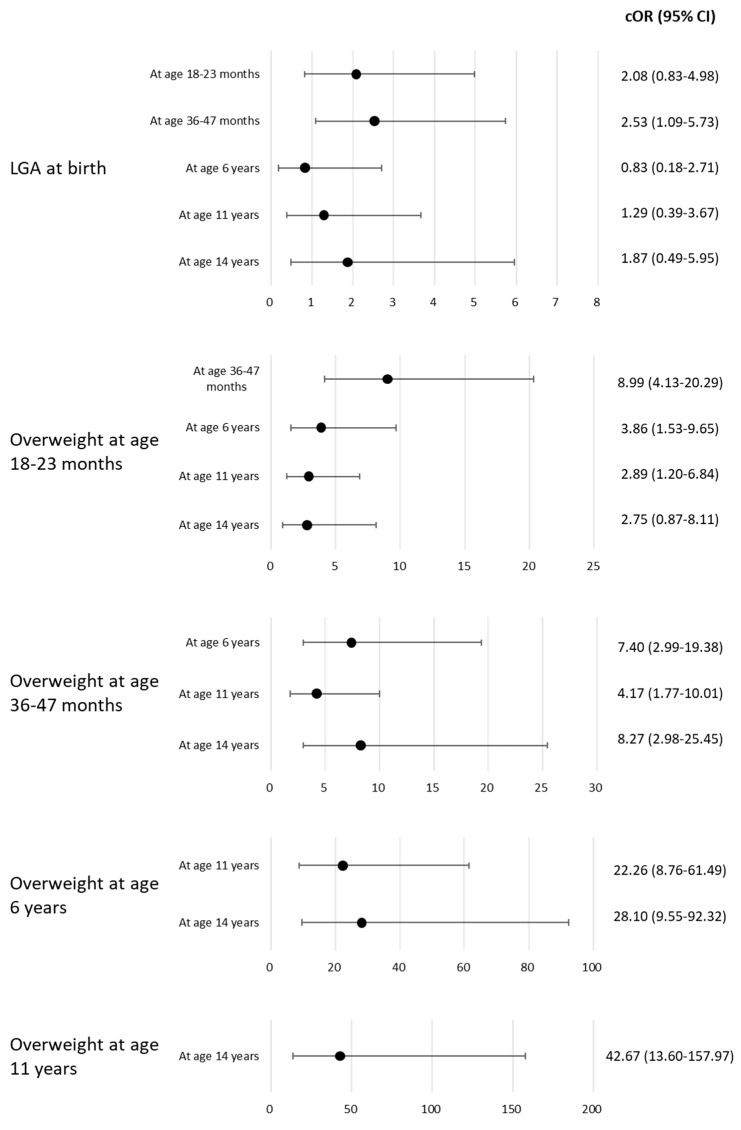
OR and 95% CI from logistic regression analysis of the association between LGA at birth or overweight at any time point and overweight at subsequent time points. cOR, crude odds ratio; CI, confidence interval; LGA, large-for-gestational age.

**Table 1 children-10-00909-t001:** Characteristics of children.

Variables	n	Overall, n = 246 ^1^	Man, n = 119 ^1^	Woman, n = 127 ^1^	*p*-Value ^2^
Length at birth (cm)	227	49.54 (1.89)	49.69 (1.90)	49.39 (1.88)	0.3
Missing, n		19	10	9	
Body weight at birth (g)	227	3,111 (356)	3,142 (336)	3,081 (372)	0.2
Missing, n		19	10	9	
Gestational age (weeks)	227	39.56 (1.28)	39.60 (1.06)	39.51 (1.45)	0.7
Missing, n		19	10	9	
Body types at birth	227				>0.9
LGA, n (%)		36 (15.9%)	16 (14.7%)	20 (16.9%)	
AGA, n (%)		183 (80.6%)	89 (81.7%)	94 (79.7%)	
SGA, n (%)		8 (3.5%)	4 (3.7%)	4 (3.4%)	
Missing, n		19	10	9	
Height at age 18–24 months (cm)	191	81.08 (3.01)	81.62 (2.81)	80.56 (3.12)	0.015 *
Missing, n		55	26	29	
Body weight at age 18–24 months (kg)	191	10.74 (1.20)	11.04 (1.21)	10.45 (1.11)	0.002 *
Missing, n		55	26	29	
BMI at age 18–24 months (kg/m^2^)	191	16.31 (1.20)	16.55 (1.29)	16.07 (1.07)	0.009 *
Missing, n		55	26	29	
Body types at age 18–24 months	191				0.6
Normal weight, n (%)		155 (81.2%)	77 (82.8%)	78 (79.6%)	
Overweight, n (%)		36 (18.8%)	16 (17.2%)	20 (20.4%)	
Missing, n		55	26	29	
Height at age 36–47 months (cm)	211	96.8 (3.9)	96.9 (3.7)	96.7 (4.0)	>0.9
Missing, n		35	16	19	
Body weight at age 36–47 months (kg)	211	14.93 (1.76)	14.91 (1.56)	14.94 (1.94)	>0.9
Missing, n		35	16	19	
BMI at age 36–47 months (kg/m^2^)	211	15.90 (1.17)	15.85 (0.99)	15.94 (1.32)	0.7
Missing, n		35	16	19	
Body types at age 36–47 months	211				>0.9
Normal weight, n (%)		160 (75.8%)	78 (75.7%)	82 (75.9%)	
Overweight, n (%)		51 (24.2%)	25 (24.3%)	26 (24.1%)	
Missing, n		35	16	19	
Height at age 6 years (cm)	179	116.7 (5.2)	117.0 (5.1)	116.3 (5.3)	0.6
Missing, n		67	28	39	
Body weight at age 6 years (kg)	179	22.2 (4.2)	22.1 (4.0)	22.3 (4.4)	0.6
Missing, n		67	28	39	
BMI at age 6 years (kg/m^2^)	179	16.20 (2.05)	16.01 (1.86)	16.38 (2.22)	0.4
Missing, n		67	28	39	
Body types at age 6 years	179				0.089
Normal weight		149 (83.2%)	80 (87.9%)	69 (78.4%)	
Overweight		30 (16.8%)	11 (12.1%)	19 (21.6%)	
Missing, n		67	28	39	
Height at age 11 years (cm)	174	146 (7)	145 (8)	148 (7)	0.005 *
Missing, n		72	30	42	
Body weight at age 11 years (kg)	174	41 (9)	40 (10)	42 (8)	0.026 *
Missing, n		72	30	42	
BMI at age 11 years (kg/m^2^)	174	18.9 (3.2)	18.7 (3.3)	19.1 (3.1)	0.2
Missing, n		72	30	42	
Body types at age 11 years	174				0.4
Normal weight		139 (79.9%)	69 (77.5%)	70 (82.4%)	
Overweight		35 (20.1%)	20 (22.5%)	15 (17.6%)	
Missing, n		72	30	42	
Height at age 14 years (cm)	176	161 (7)	166 (6)	157 (6)	<0.001 *
Missing, n		70	32	38	
Body weight at age 14 years (kg)	176	54 (10)	56 (12)	52 (8)	0.037 *
Missing, n		70	32	38	
BMI at age 14 years (kg/m^2^)	176	20.9 (3.5)	20.3 (3.7)	21.4 (3.2)	0.001 *
Missing, n		70	32	38	
Body types at age 14 years	176				0.15
Normal weight, n (%)		151 (85.8%)	78 (89.7%)	73 (82.0%)	
Overweight, n (%)		25 (14.2%)	9 (10.3%)	16 (18.0%)	
Missing, n		70	32	38	

^1^ Mean (SD). ^2^ Wilcoxon rank sum test; Pearson’s Chi-squared test; Fisher’s exact test. * Denotes significance, *p*-value < 0.05. SD, standard deviation; BMI, body mass index; LGA, large for gestational age; AGA, appropriate for gestational age; SGA, small for gestational age.

**Table 2 children-10-00909-t002:** Characteristics of mothers.

Variables	n	Overall, n = 246 ^1^	Man, n = 119 ^1^	Woman, n = 127 ^1^	*p*-Value ^2^
Maternal age at the time of childbirth (years)	227	25.0 (3.5)	24.8 (3.4)	25.1 (3.7)	0.7
Missing, n		19	9	10	
Maternal parity	229				0.5
Multipara, n (%)		45 (19.7%)	20 (18.0%)	25 (21.2%)	
Primipara, n (%)		184 (80.3%)	91 (82.0%)	93 (78.8%)	
Missing, n		17	8	9	
Maternal BMI when pregnancy was detected (kg/m^2^)	178	20.70 (2.94)	20.57 (2.97)	20.83 (2.93)	0.5
Missing, n		68	35	33	
Alcohol consumption when pregnancy was detected	226				0.3
Yes, n (%)		17 (7.5%)	9 (8.3%)	8 (6.8%)	
No, n (%)		168 (74.3%)	75 (69.4%)	93 (78.8%)	
Unknown, n (%)		41 (18.1%)	24 (22.2%)	17 (14.4%)	
Missing, n		20	11	9	
Smoking when pregnancy was detected	227				0.2
Yes, n (%)		18 (7.9%)	11 (10.1%)	7 (5.9%)	
No, n (%)		172 (75.8%)	77 (70.6%)	95 (80.5%)	
Unknown		37 (16.3%)	21 (19.3%)	16 (13.6%)	
Missing, n		19	10	9	

^1^ Mean (SD). ^2^ Wilcoxon rank sum test; Pearson’s Chi-squared test; Fisher’s exact test. SD, standard deviation; BMI, body mass index.

**Table 3 children-10-00909-t003:** OR and 95% CI from logistic regression analysis of the association between body types at any time point and that of subsequent time points.

Outcome	At Age 18–23 Months		At Age 36–47 Months		At Age 6 Years		At Age 11 Years		At Age 14 Years	
Exposure	n (%)	aOR * (95% CI)	n (%)	OR * (95% CI)	n (%)	OR * (95% CI)	n (%)	OR * (95% CI)	n (%)	OR *(95% CI)
At birth	120		124		92		86		93	
AGA	97	1.00	101	1.00	78	1.00	72	1.00	77	1.00
	(80.8)	(Ref.)	(81.5)	(Ref.)	(84.8)	(Ref.)	(83.7)	(Ref.)	(82.8)	(Ref.)
LGA	19	1.26	18	2.59	11	0.00	11	0.45	12	5.38
	(15.8)	(0.32–4.22)	(14.5)	(0.80–7.98)	(12.0)	(N.A. −3.06 × 1047)	(12.8)	(0.02–3.35)	(82.8)	(0.17–115.71)
SGA	4	2.40 × 10^−8^	5	1.29	3	1.05 × 10^−7^	3	5.43	4	103.64
	(3.3)	(N.A. −1.13 × 10,148)	(4.0)	(0.06–9.76)	(3.3)	(N.A. −2.41 × 10,184)	(3.5)	(0.20–87.69)	(4.3)	(1.89–10,610.63)
At age 18–23 months			119		95		88		91	
Normal weight			93	1.00	73	1.00	66	1.00	73	1.00
			(78.2)	(Ref.)	(76.8)	(Ref.)	(75.0)	(Ref.)	(80.2)	(Ref.)
Overweight			26	13.42	22	6.94	22	5.22	18	1.61
			(21.8)	(4.46–45.42)	(23.2)	(1.64–33.46)	(25.0)	(1.25–24.79)	(19.8)	(0.14–14.96)
At age 36–47 months					89		85		91	
Normal weight					69	1.00	64	1.00	71	1.00
					(77.5)	(Ref.)	(75.3)	(Ref.)	(78.0)	(Ref.)
Overweight					20	9.55	21	3.22	20	0.29
					(22.5)	(1.85–61.41)	(24.7)	(0.70–15.14)	(22.0)	(0.01–4.47)
At age 6 years							91		80	
Normal weight							80	1.00	74	1.00
							(87.9)	(Ref.)	(92.5)	(Ref.)
Overweight							11	1021.24	6	13.15
At age 11 years							(12.1)	(48.96–81,487.89)	79	(0.97–261.80)
Normal weight									69	1.00
									(87.3)	(Ref.)
Overweight									10	60.46
									(12.7)	(2.92–5001.37)

* Adjusted for gender, maternal age at childbirth, maternal parity, maternal BMI when determining pregnancy, maternal drinking status, pregnancy confirmation, and smoking status at pregnancy confirmation. aOR, adjusted odds ratio; CI, confidence interval; Ref., reference value; N.A., not available; LGA, large for gestational age; AGA, appropriate for gestational age; SGA, small for gestational age.

## Data Availability

A biobank is being constructed based on the TMM BirThree Cohort Study. The full baseline data have been distributed to researchers who have been approved by the Sample and Data Access Committee of the Biobank since 2017.

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
