# Peer review of "Association between being Overweight in Young Childhood and during School Age and Puberty"

_children, 2023, doi:10.3390/children10050909_

Round 1

Reviewer 1 Report

Dear Sir / Madam,

Please find below my comments and suggestions following the review of the study titled ‘Associations between overweight in infancy and overweight in school age and puberty’.

The topic chosen by the authors is interesting and I appreciate the efforts of the authors to link data with that being collected by other agencies so as to avoid duplication of efforts and wastage of resources. However, I believe that the data needs to be better handled. The Introduction and Discussion sections do not do justice to the objectives and results, respectively. The results need to be presented in a manner that makes them easy to comprehend. There seem to be a few grammatical errors in the manuscript. Apart from the modification suggested below, I would recommend to the authors to focus on data visualization and lucid presentation of their ideas.

1.      Abstract:

·         Line 22-28 – “We comprehensively examined the association ….. school physical examination information.” This is a very long sentence and difficult to read. It would help if the authors split this sentence into 2 sentences. Also, there seems to be an error or some information missing in line 27 which makes it difficult to ascertain the meaning of that portion.

·         Line 28-29 – Please replace the words ‘higher percentage’ with ‘greater risk’.

·         Line 30 – Instead of 1-year-old checkup, please write at the age of 1 year.

2.      Introduction:

·         The authors need to elaborate their reasoning behind studying or linking maternal health to overweight. This has not been touched upon at all in the Introduction and the concept suddenly appears in the last paragraph of the Introduction (Line 65).

3.      Materials and Methods:

·         Line 82 “They were newborn siblings or extended family members.” The meaning of this sentence is unclear.

·         Please provide details of sample size calculation.

4.      Results:

·         Line 136-138 The meaning of this sentence is unclear.

·         Table 1 – It is length at birth and not height at birth.

·         Table 1 – Monopara is not a word. It is primipara.

·         Please change the names of the following variables in Table 1 – Maternal Characteristics section

·         Maternal BMI when find out pregnancy – Maternal BMI when pregnancy was detected

·         Drinking alcohol when find out – Alcohol Consumption when pregnancy was detected

·         Smoking when find out pregnancy – Smoking when pregnancy was detected

·         Table 2 needs to be restructured so as to improve comprehension of the results.

5.      Discussion:

·         Line 219 “Infant and school-age children” Please revise as ‘Infancy and school age”.

Author Response

Thank you very much for providing important comments. We are thankful for the time and energy you expended. Our responses to the referees’ comments are as follow:

  1. Abstract:
  • Line 22-28 – “We comprehensively examined the association ….. school physical examination information.” This is a very long sentence and difficult to read. It would help if the authors split this sentence into 2 sentences. Also, there seems to be an error or some information missing in line 27 which makes it difficult to ascertain the meaning of that portion.

→ I also revised the sentence and divided it into two sentences as follow.

“Data from maternal and child health handbooks, baby health checkup information, and school physical examination information of the participants in the Birth and Three-Generation Cohort Study were linked. Then, we comprehensively examined the association between body type and body weight at different time intervals at birth, at 1.5, 3.5, 6, 11, and 14 years of age using a multivariate regression model adjusted for sex, maternal age at childbirth, maternal parity, and maternal body mass index, drinking status, and smoking status at the time they find out they are pregnant.”

  • Line 28-29 – Please replace the words ‘higher percentage’ with ‘greater risk’.

→ I finished editing it.

  • Line 30 – Instead of 1-year-old checkup, please write at the age of 1 year.

→ I finished editing it.

  1. Introduction:
  • The authors need to elaborate their reasoning behind studying or linking maternal health to overweight. This has not been touched upon at all in the Introduction and the concept suddenly appears in the last paragraph of the Introduction (Line 65).

→ We quoted a study regarding maternal health and overweight of their child in our Introduction and added a rationale that we have used data on maternal BMI, alcohol consumption, and smoking as follow (Line 70-75).

“This highlights the need for interventions to prevent obesity from childhood. Furthermore, maternal body mass index (BMI) [13], drinking alcohol [14] and smoking habits [15] have been reported to be risk factors for obesity in children. Collectively, tracking data during the child's growing up process from prenatal are required.”

  1. Materials and Methods:
  • Line 82 “They were newborn siblings or extended family members.” The meaning of this sentence is unclear.

→ (Line 82) The sentence "They were newborn siblings or extended family members." was deleted because it was considered misleading.

  • Please provide details of sample size calculation.

→ The sample size calculation method is described in our Results section as follow.

Figure 1. Flow chart of participants in this study

*height, body weight, and measurement date

However, would it be better to describe it in more detail?

  1. Results:
  • Line 136-138 The meaning of this sentence is unclear.

→ I corrected that writing a little.

  • Table 1 – It is length at birth and not height at birth.

→ I finished editing it.

  • Table 1 – Monopara is not a word. It is primipara.

→ I finished editing it.

  • Please change the names of the following variables in Table 1 – Maternal Characteristics section

- Maternal BMI when find out pregnancy – Maternal BMI when pregnancy was detected

- Drinking alcohol when find out – Alcohol Consumption when pregnancy was detected

- Smoking when find out pregnancy – Smoking when pregnancy was detected

→ I finished editing the names of the following variables in Table 1.

  • Table 2 needs to be restructured so as to improve comprehension of the results.

→ Thank you for your valuable comment. We have restructured Table 2; We have utilized crude data in Figure 2 and created Table 2 only for adjusted data.

  1. Discussion:
  • Line 219 “Infant and school-age children” Please revise as ‘Infancy and school age”.

→ I finished editing it.

Reviewer 2 Report

I reviewed the interesting article by Shinoda et al., titled " Associations between overweight in infancy and overweight in school age and puberty" submitted to the prestigious journal CHILDREN. The study presented in the manuscript effectively tends to assess the correlation between being overweight in infancy and in school age and puberty

The strength of the work is that it is concise and thorough, with an illuminating introduction that accentuates the theme being discussed. It is recommended that the work may be published in the journal after a few corrections and suggestions are incorporated.

The author can consider discussing a few points from the studies below.

1.     The authors may want to consider adding a figure to shed light on the central theme of the paper.

2.     Authors are suggested to discuss more about cancer and childhood obesity. Please discuss from a recent article published in MDPI Life https://www.mdpi.com/2075-1729/12/10/1673

3.     To emphasize the issue being discussed, the authors have the option to incorporate statistical information regarding the global prevalence and impact of obesity. The studies below can be considered for reference;
https://www.thelancet.com/journals/eclinm/article/PIIS2589-5370(23)00027-5/fulltext
https://www.who.int/news-room/fact-sheets/detail/obesity-and-overweight

4.     To broaden the scope of the study, the authors may consider including additional studies from diverse ethnic cohorts to fortify the argument. The following papers can be considered;
https://www.ncbi.nlm.nih.gov/pmc/articles/PMC8782254/
https://www.nature.com/articles/ijo2013115
https://www.ncbi.nlm.nih.gov/pmc/articles/PMC1261184/
https://jhpn.biomedcentral.com/articles/10.1186/s41043-015-0012-2
https://www.nejm.org/doi/full/10.1056/nejmoa1703860 https://www.ncbi.nlm.nih.gov/pmc/articles/PMC6702696/

I believe the paper would be an excellent contribution to the literature after the revisions as suggested above.

Author Response

  1. The authors may want to consider adding a figure to shed light on the central theme of the paper.

→ According to your comment, we have added Figure 2, which illustrated the crude association of LGA at birth or overweight at any time point and overweight of subsequent time point.

  1. Authors are suggested to discuss more about cancer and childhood obesity. Please discuss from a recent article published in MDPI Life https://www.mdpi.com/2075-1729/12/10/1673

→ Thank you for your valuable comment. We have added to our discussion of the association between cancer and childhood obesity with reference to the literature you have provided as follow (Line 230-233).

“Besides that, obesity has been shown to be associated with onset of cancer and is one of the potential causes behind the onset of the central nervous system tumors, including glioblastoma multiforme [27].”

  1. To emphasize the issue being discussed, the authors have the option to incorporate statistical information regarding the global prevalence and impact of obesity. The studies below can be considered for reference; https://www.thelancet.com/journals/eclinm/article/PIIS2589-5370(23)00027-5/fulltext https://www.who.int/news-room/fact-sheets/detail/obesity-and-overweight

→ Thank you for your valuable comment. We have added the global prevalence and impact of obesity to the Introduction with reference to the literature you provided as follow (Line 52-55 and 57-60).

“The World Health Organization reports that the worldwide prevalence of overweight and obesity in childhood and puberty aged 5-19 has risen dramatically from 4% in 1975 to over 18% in 2016, and this rise has occurred in both sexes [2].”

“According to a study using data from the 2019 Global Burden of Disease study, obesity-related disability-adjusted life years rose 0.48% annually from 2000 to 2019 and are predicted to increase 39.8% from 2020 to 2030 [8].”

  1. To broaden the scope of the study, the authors may consider including additional studies from diverse ethnic cohorts to fortify the argument. The following papers can be considered; https://www.ncbi.nlm.nih.gov/pmc/articles/PMC8782254/

https://www.nature.com/articles/ijo2013115

https://www.ncbi.nlm.nih.gov/pmc/articles/PMC1261184/

https://jhpn.biomedcentral.com/articles/10.1186/s41043-015-0012-2

https://www.nejm.org/doi/full/10.1056/nejmoa1703860

https://www.ncbi.nlm.nih.gov/pmc/articles/PMC6702696/

I believe the paper would be an excellent contribution to the literature after the revisions as suggested above.

→ Thank you very much for providing us with the literature. I have added a discussion based on some of them as follow (Line 255-263).

“Furthermore, obesity is not just a problem for children and adults. With the increasing number of elderly people in the world, obesity is a powerful risk factor affecting health in old age. Especially, dementia, including alzheimer's disease, is one of the factors that greatly reduces activities of daily living in old age, though there is fragmented understanding of the mechanisms connecting obesity and ageing [35].  Therefore, life course data, including not only childhood but also adulthood and old age will serve to help proper weight management and obesity prevention in the future.”

Reviewer 3 Report

Comments and suggestions to authors:

·         Pay attention to formatting, for example, nouns in the title should be capitalized; the source list is not formatted according to the guidelines. Sometimes unreasonably old references.

Formatting examples:

Author 1, A.B.; Author 2, C.D. Title of the article. Abbreviated Journal Name Year, Volume, page range.

Gilley, P.M.; Sharma, A.; Dorman, M.F. Cortical Reorganization in children with cochlear implants. Brain Res. 2008, 1239, 56–65.

Uhler, K.M.; Hunter, S.K.; Tierney, E.; Gilley, P.M. The relationship between mismatch response and the acoustic change complex in normal hearing infants. Clin. Neurophysiol. 2018, 129, 1148–1160.

·         I'm not good at English wording, but sometimes I couldn't understand the meaning of the sentences. Including, for example, the 1st sentence of the abstract. Is this the aim of the study? Replace "sex" with "gender" throughout the manuscript. Line 30 remains unclear. The first sentences of the introduction essentially repeat the same idea. The last sentence of the first paragraph (line 46) is meaningless and has no reference. The literature review needs substantive revision and rewriting according to the topic. Sentence structure and phrasing should be reviewed throughout the manuscript (eg lines 145-146), what does the author want to say?

·         The methodology could also provide an overview of the permission of the ethics committee and the obtaining of informed consents from the subjects. Was the permission to process personal data asked only from the parents or also from the children? Is this data really based on studies that were approved by the Ethics Committee in 2013?

·         On what basis do you claim that "body type" might be defined using sex, height, body weight, measurement dates, date of birth, and gestational age? In Figure 1, you instead state that "body type at birth - *height, body weight, and measurement date" (line 127).

·         The tables are long and completely incomprehensible. Think about what data and how to present it.

·         Sometimes they talk about a specific age, then again "childhood, school age and puberty". So that the ages are consistent throughout the text, because the exact age range of "school age" remains unclear to the reader and children do not have puberty at the same time.

·         It remains unclear what the true value of this data is. The authors describe that it is important to "investigate" the obesity of newborns in the future, but what does this so-called investigation actually provide? I would like a more substantive analysis!

Best wishes

Author Response

Thank you very much for providing important comments. We are thankful for the time and energy you expended. Our responses to the referees’ comments are as follow:

  • Pay attention to formatting, for example, nouns in the title should be capitalized; the source list is not formatted according to the guidelines. Sometimes unreasonably old references.

Formatting examples:

Author 1, A.B.; Author 2, C.D. Title of the article. Abbreviated Journal Name Year, Volume, page range.

Gilley, P.M.; Sharma, A.; Dorman, M.F. Cortical Reorganization in children with cochlear implants. Brain Res. 2008, 1239, 56–65.

Uhler, K.M.; Hunter, S.K.; Tierney, E.; Gilley, P.M. The relationship between mismatch response and the acoustic change complex in normal hearing infants. Clin. Neurophysiol. 2018, 129, 1148–1160.

→ Title and references formats have been corrected.

  • I'm not good at English wording, but sometimes I couldn't understand the meaning of the sentences. Including, for example, the 1st sentence of the abstract. Is this the aim of the study?

→ You are correct. The 1st sentence of the abstract is the aim of this study.

Replace "sex" with "gender" throughout the manuscript.

→ All "gender" has been corrected to "sex".

Line 30 remains unclear.

→ Line 30 has been revised to the following text (line 38-41).

“In particular, overweight at the 1-year-old checkup was associated with overweight at 3.5 years (adjusted odds ratio (aOR), 13.42; 95% confidence interval (CI), 4.46-45.42), 6 years (aOR, 6.94; 95% CI, 1.64-33.46) and 11 years (aOR, 5.22; 95% CI, 1.25-24.79) of age.”

The first sentences of the introduction essentially repeat the same idea. The last sentence of the first paragraph (line 46) is meaningless and has no reference.

→ According to your comments, the following sentence in line 46 and has been removed.

“Obesity is an important public health concern in countries where lifestyle-related diseases are a significant health issue.”

The literature review needs substantive revision and rewriting according to the topic. Sentence structure and phrasing should be reviewed throughout the manuscript (eg lines 145-146), what does the author want to say?

→ I finished editing it.

  • The methodology could also provide an overview of the permission of the ethics committee and the obtaining of informed consents from the subjects.

→ The following text was added at the end of 2.1 Study design in 2. Materials & Methods (line 110-113).
“The study protocol was approved by the ethics committee of the Tohoku Medical Megabank Organization, and it is conformed to the provisions of the Declaration of Helsinki. Written informed consent was obtained from all participants.”

Was the permission to process personal data asked only from the parents or also from the children?

→ Our response to your question is as follows (line 98-100).
“We received signed informed consent from guardians who wished to participate, and parental consent for children under 10 years of age, and their own assent and parental consent for children 10-14 years of age.”

Is this data really based on studies that were approved by the Ethics Committee in 2013?

→ This prospective study was approved by the Ethics Committee in 2013.

  • On what basis do you claim that "body type" might be defined using sex, height, body weight, measurement dates, date of birth, and gestational age? In Figure 1, you instead state that "body type at birth - *height, body weight, and measurement date" (line 127).

→ Sex, date of birth, and gestational age were added to the description of body type at birth in Figure 1.

  • The tables are long and completely incomprehensible. Think about what data and how to present it.

→ I finished editing it.

  • Sometimes they talk about a specific age, then again "childhood, school age and puberty". So that the ages are consistent throughout the text, because the exact age range of "school age" remains unclear to the reader and children do not have puberty at the same time.

→ In this study, the age and timing coincidences are as follows. And the following text was added to “2.1 Study design” in Materials & Methods (line 108-110).
“In this study, 1.5 and 3.5 years old were defined as "infancy," 6 and 11 years old as "school age," and 14 years old as "puberty.”

  • It remains unclear what the true value of this data is. The authors describe that it is important to "investigate" the obesity of newborns in the future, but what does this so-called investigation actually provide? I would like a more substantive analysis!

→ I would like to make this an issue for future discussion. Also, add the following text to the "Discussion" section (line 286-288).
“It is important to follow-up obese children and to investigate their prognosis in the future.”

Round 2

Reviewer 1 Report

The authors have taken considerable efforts to modify the manuscript as per my suggestions. Yet a number of grammatical errors are found throughout the manuscript.

For example,

Line 62-63 "And obese children are associated with an increased risk of morbidities, including diabetes, coronary heart disease (CHD), and a range of cancers"

Line 176-177 "A slightly younger of boys maternal age at the time of childbirth (boys, 24.8±3.4 years vs. girls, 25.1±3.7 years) than girls."

The authors are requested to make the required corrections and get the manuscript edited by a Native English speaker.

Author Response

Thank you very much for providing important comments. We are thankful for the time and effort you expended. Our responses to the referees’ comments are as follow:

The authors have taken considerable efforts to modify the manuscript as per my suggestions. Yet a number of grammatical errors are found throughout the manuscript.

For example,

Line 62-63 "And obese children are associated with an increased risk of morbidities, including diabetes, coronary heart disease (CHD), and a range of cancers"

Line 176-177 "A slightly younger of boys maternal age at the time of childbirth (boys, 24.8±3.4 years vs. girls, 25.1±3.7 years) than girls."

The authors are requested to make the required corrections and get the manuscript edited by a Native English speaker.

⇒ Thank you for your comment.

Although our manuscript was English proofread by Editage, the revised version of our manuscript was English proofread again.

I’ve attached the Editing Certificate.

Reviewer 3 Report

Corrections have been made by the authors.

However, I pay attention to the spelling throughout the work. However, the word "sex" must be replaced with the word "gender", not the other way around. It remains unclear to the reader what exactly "school age" means. The reference list is not completely compiled according to the guide. The authors repeatedly use the form "we" and the work does not leave a scientific impression in its content or form.

Author Response

Thank you very much for providing important comments. We are thankful for the time and effort you expended. Our responses to the referees’ comments are as follow:

Corrections have been made by the authors.

However, I pay attention to the spelling throughout the work. However, the word "sex" must be replaced with the word "gender", not the other way around.

⇒ Thank you for your comment.

   We have been replaced the word "sex" with the word "gender".

It remains unclear to the reader what exactly "school age" means.

⇒ In this study, the timing "school age" represents the age "6 and 11 years old". The following text was added to “2.1 Study design” in Materials & Methods. (Line 102-104)

   In this study, 1.5 to 3.5 years of age was defined as "infancy”, 6 to 11 years old as "school age”, and 14 years old as "puberty”.

The reference list is not completely compiled according to the guide.

⇒ We have reviewed the regulations of the journal again and modified our reference list.

The authors repeatedly use the form "we" and the work does not leave a scientific impression in its content or form.

⇒ According to your comments, we have checked the expression of "we" and deleted the form "we believe" as follows.

Therefore, we believe our finding suggests that interventions for obesity, not only at school age but also during infancy or earlier, are important for the prevention of obesity in puberty and adulthood. (Line 231-233)

Considering the results of this study, we believe that linking the life course data is important for formulating measures to prevent obesity during puberty. (Line 247-248)

Based on the above results, we believe that the prevention of being overweight at school age and puberty requires avoiding being overweight in infancy, especially during the 1.5-year-old checkup period. (Line 293-295)